# Cardiovascular Risks in Korean Patients with Gout: Analysis Using a National Health Insurance Service Database

**DOI:** 10.3390/jcm11082124

**Published:** 2022-04-11

**Authors:** Ki Won Moon, Min Jung Kim, In Ah Choi, Kichul Shin

**Affiliations:** 1Division of Rheumatology, Department of Internal Medicine, Kangwon National University School of Medicine, Chuncheon 24289, Korea; kiwonmoon@kangwon.ac.kr; 2Division of Rheumatology, Department of Internal Medicine, Seoul Metropolitan Government-Seoul National University Boramae Medical Center, Seoul 07061, Korea; fairytaie@naver.com; 3Division of Rheumatology, Department of Internal Medicine, College of Medicine, Chungbuk National University, Cheongju 28644, Korea; sylph014@hanmail.net

**Keywords:** gout, urate lowering therapy, cerebrovascular disease

## Abstract

Although several epidemiologic studies have shown the association between gout and cardiovascular outcomes, specific risk factors for developing cardiovascular diseases in Asian patients with gout are undisclosed. Thus, the purpose of this study was to investigate risks of cardiovascular outcomes and its related factors in Korean patients with gout. This retrospective clinical study used sampled cohort data from the National Health Insurance Service in Korea. Patients with gout were defined as subjects enlisted with an ICD-10 code (M10). Control patients were selected by frequency matching for age, sex, and index year. Primary outcomes included ischemic heart disease (IHD), congestive heart failure, cerebrovascular disease (CVD), or transient ischemic attack. We calculated the hazard ratio (HR) using Cox regression, adjusting potential confounders including age, sex, lifestyle habits, laboratory results, and medication. We identified 3306 patients with gout and an equal number of matched controls. Multivariate Cox regression analysis showed that gout patients had increased risks of IHD (HR: 1.860, 95% CI: 1.446–2.392), acute myocardial infarction (HR: 3.246, 95% CI: 1.460–7.217), and CVD (HR: 1.552, 95% CI: 1.177–2.036). Old age, current smoking, frequent alcohol intake, high low-density lipoprotein, and diabetes mellitus increased the risk of cardiovascular outcomes, yet hypouricemic agents decreased the risk of cerebrovascular diseases. Our data corroborate that it is crucial to identify and manage traditional cardiovascular risk factors alongside lowering urate levels in patients with gout.

## 1. Introduction

Gout is associated with an increased risk for cardiovascular comorbidities and premature mortality [1]. According to a recent population-based study, there has been no discernable improvement of the risk of premature mortality among gout patients in recent years [2]. Several epidemiologic studies have supported an association between gout and increased risk for cardiovascular diseases such as ischemic heart disease (IHD), acute myocardial infarction (AMI), cerebrovascular disease (CVD), congestive heart failure (CHF), and transient ischemic attack (TIA) [3,4]. In addition, some studies have reported that gout is associated with a higher risk of death from all causes and cardiovascular diseases [5,6]. However, there has been no study on cardiovascular risks or mortality in Korean patients with gout using large scale data. In addition, evidence about which risk factors contribute to cardiovascular risks and mortality in patients with gout remains insufficient. Therefore, the objective of this study was to evaluate the risk of subsets of cardiovascular diseases and its mortality and to identify specific risk factors in Korean patients with gout utilizing a National Health Insurance database.

## 2. Materials and Methods

### 2.1. Data Sources

This study analyzed data from the National Health Insurance Service-National Sample Cohort (NHIS-NSC). The NHIS-NSC is a population-based database established by the National Health Insurance Service (NHIS) in South Korea [7]. The NHIS is a single player health insurance system that covers the entire South Korean population. The NHIS-NSC was instituted in 2002. It randomly selected about 1 million participants, comprising 2.2% of the total eligible population. These participants were followed until 2015. The NHIS-NSC contains longitudinal patient data including demographics, diagnosis code, procedures, and prescription record. It also includes major health examination results and information about lifestyles and behaviors obtained from questionnaires. Protocols of this study were reviewed and approved by the Institutional Review Board of Kangwon National University Hospital (KNUH-2022-01-014).

### 2.2. Study Population

Using the NHIS-NSC, we conducted a retrospective study on gout patients. Gout patients were defined as those aged ≥20 years who were diagnosed as gout between 1 January 2009 and 31 December 2010. Control patients were selected (1:1 ratio) by frequency matching for age, sex, and index year, excluding subjects who were diagnosed with gout at least once during the study period. Diagnosis of gout was defined when the 10th edition of International Classification of Diseases (ICD-10) code (M10) was registered for more than two occasions. The index date was the date when the M10 code was first registered. To ensure that only new cases were enrolled, patients diagnosed with gout in 2008 were excluded. The NHIS-NSC database contains major health examination results including lifestyles, health behavior, and laboratory findings. We used health examination results between 2009 and 2010 because major changes in health examination questionnaires and laboratory items were made in 2009. Several important laboratory results including creatinine and lipid profiles were unavailable before 2009. We excluded patients who had no health examination or laboratory results. We also excluded patients diagnosed with cardiovascular diseases before the index date.

### 2.3. Definitions of Outcomes

The primary outcome was the first event of cardiovascular diseases, such as ischemic heart diseases (IHD) including acute myocardial infarct (AMI), congestive heart failure (CHF), cerebrovascular diseases (CVD) including cerebral infarction and cerebral hemorrhage, and transient ischemic attacks (TIA). They were identified by two or more diagnosis of IHD (ICD-10 codes I20 to I25), AMI (ICD-10 codes I21 to I23), CHF (ICD-10 code I50), CVD (ICD-10 codes I60 to I69), cerebral infarction (ICD-10 code I63), cerebral hemorrhage (ICD-10 codes I60 to I62), and TIA (ICD-10 code G45). Secondary outcomes were all-cause mortality and cardiovascular mortality. Dates and causes of death were obtained from records included in the dataset of NHIS-NSC. A full year after the index was used as the wash-out period. Thus, newly diagnosed events were captured during follow-up visits between 2010 and 2015. 

### 2.4. Assessment of Covariates

We assessed certain variables associated with cardiovascular outcomes from the year prior to the index date. These variables were: age, sex, body mass index (BMI), lifestyle habits (smoking, alcohol intake, and exercise) laboratory findings (fasting glucose, total cholesterol, triglyceride, high density lipoprotein (HDL), low density lipoprotein (LDL), creatinine, and glomerular filtration rate (GFR)), comorbidities (hypertension, diabetes mellitus, hyperlipidemia, and chronic kidney disease), medication (diuretics, beta-blocker, calcium channel blocker, angiotensin converting enzyme inhibitor (ACEI), angiotensin receptor blocker (ARB), non-steroidal anti-inflammatory drug (NSAID), cyclooxygenase (cox)-2 inhibitor, aspirin, steroid, statin, colchicine, and hypouricemic agents). Comorbidities were defined as hypertension (ICD-10 codes I10 to I15), diabetes (ICD-10 codes E10 to E14), hyperlipidemia (ICD-10 code E78), and chronic kidney disease (ICD-10 codes N17 to N19). Hypouricemic agents included allopurinol, febuxostat, and benzbromarone. 

### 2.5. Statistical Analysis

Frequencies and percentages were calculated for categorical variables and means with standard deviations were calculated for continuous variables. Chi-square and independent *t*-tests were used to analyze categorical variables and continuous variables, respectively. Multivariate Cox proportional hazards regression analysis was used to determine hazard ratio (HR) for developing cardiovascular events after adjusting confounding variables. Results are shown as adjusted HR and 95% confidence intervals (CI). *P*-values less than 0.05 were considered statistically significant. All statistical analyses were conducted using SAS software version 9.4 (SAS Institute, Cary, NC, USA). 

## 3. Results

This study included 3306 gout patients and an equal number of matched controls. Baseline characteristics and outcome events in gout and control patients are listed in Table 1 BMI, lifestyle habits, laboratory findings, comorbidities, medication, and cardiovascular outcome events were significantly different between gout patients and matched controls. A multivariate Cox regression analysis revealed that gout patients showed increased risks of IHD (HR: 1.860, 95% CI: 1.446–2.392), AMI (HR: 3.246, 95% CI: 1.460–7.217), and CVD (HR: 1.552, 95% CI: 1.177–2.036) in comparison to control patients after adjusting for age, sex, lifestyle habits, laboratory results, comorbidities, and medication (Table 2). However, there was no significant difference in the risk of CHF, cerebral infarction, cerebral hemorrhage, TIA, all-cause mortality, or cardiovascular mortality between the two. Additional Cox regression analysis was performed for three cardiovascular outcomes (IHD, AMI, and CVD) in gout patients (Table 3). Old age was a risk factor for all three cardiovascular outcomes. Current smokers had a significant higher risk for IHD (HR: 1.315, 95% CI: 1.021–1.695). High LDL and diabetes were significant risk factors for AMI (HR: 3.008, 95% CI: 1.368–6.618, HR: 2.070, 95% CI: 1.118–3.833, respectively). Frequent alcohol intake and diabetes were significant risk factors for CVD (HR: 1.636 95% CI: 1.069–2.502 and HR: 1.384, 95% CI: 1.056–1.815, respectively). Of note, taking hypouricemic agents was associated with a lower risk of CVD (HR: 0.713, 95% CI: 0.556–0.915).

## 4. Discussion

We utilized the NHIS-NSC and demonstrated that cardiovascular risks in Korean patents with gout were significantly increased, especially for IHD, AMI, and CVD. This is the first study to analyze the risks and risk factors of cardiovascular outcomes in Korean patients with gout. It has been reported that cardiovascular risks are increased in gout patients [4]. However, whether gout itself is a risk factor for cardiovascular diseases and whether the risk of cardiovascular diseases is increased by comorbidities such as hypertension, diabetes, and hyperlipidemia remain controversial. Huang et al. [8] have applied propensity score (PS) matching to answer this. Even after PS matching of comorbidities (i.e., hypertension, diabetes, hyperlipidemia, and CVD etc.), the cumulative incidence of AMI was significantly higher in gout patients than in controls. It means that gout itself might be an important risk factor for cardiovascular diseases. The mechanism by which gout increases the risk of cardiovascular disease is not fully understood yet. It has been suggested that reactive oxygen species and low-grade chronic inflammation can elevate cardiovascular risk in gout patients [9]. Reactive oxygen species induced by xanthine oxidase during uric acid production can impair endothelial nitrogen oxide production, subsequently aggravating cardiovascular diseases [10]. Uric acid is also known to promote inflammation in a variety of cells. Hyperuricemia can promote the development of atherosclerosis by regulating inflammatory pathways such as nod-like receptor protein 3 inflammasomes, macrophage M1/M2 polarization, and C-reactive protein [11]. The inflammatory activity associated with gout can be proatherogenic, promoting a prothrombotic environment that leads to acute coronary events [12]. 

It is noteworthy that baseline characteristics showed that patients with gout smoked less and consumed less alcohol. As they had a higher prevalence of hypertension, diabetes, dyslipidemia, and cardiovascular events, patients may have had chosen to maintain a healthier lifestyle and decided to quit smoking; the percentage of past smokers was higher in patients with gout. Gout patients have various comorbidities including metabolic syndrome, renal insufficiency, peripheral vascular disease, and so on [13]. They use medical services more often and take more medications than the general population [14]. Therefore, when analyzing the risk of cardiovascular outcomes, it is essential to eliminate confounding factors such as lifestyle habits, comorbidities, laboratory results, and current medications. We collected as much information as possible from the NHIS-NSC database and calculated the HR for cardiovascular outcomes considering a variety of covariates. Our results showed that risks of IHD, AMI, and CVD were increased after adjusting for the aforementioned confounding factors. The reason why risks of cardiovascular diseases other than IHD, AMI, or CVD were not significant might be due to a relatively short observation period (six years). The result could be different if the period was long enough. There was no statistically significant increase in overall mortality or cardiovascular mortality in our study. There have been conflicting results about overall and cardiovascular mortality in gout patients. Choi et al. [5] have reported that male gout patients have higher risks of overall mortality (relative risk: 1.28, 95% CI: 1.15–1.41) and cardiovascular mortality (relative risk: 1.38, 95% CI: 1.15–1.66) than men without gout from a 12-year period prospective cohort study. On the contrary, Kishnan et al. [15] have described that middle-aged men with gout have no significant risk of overall mortality (HR: 1.09, 95% CI: 1.00–1.19) or cardiovascular mortality (HR: 1.21, 95% CI: 0.99–1.49) from a 17-year follow-up study. A meta-analysis on this issue has shown that gout is associated with an increased risks of cardiovascular mortality (HR: 1.29, 95% CI: 1.14–1.44) and mortality due to coronary heart disease (HR: 1.42, 95% CI: 1.22–1.63) [6]. 

Very little information is available about which specific risk factors contribute to cardiovascular risk in gout patients. Our study showed that old age increased the risk of cardiovascular diseases, namely IHD, AMI, and CVD. Current smoking increased the risk of IHD. High LDL and diabetes increased the risk of AMI. Alcohol intake and diabetes increased the risk of CVD. Based on these results, it seems that managing traditional risk factors for cardiovascular diseases is crucial for the prognosis of gout patients. Disveld et al. [16] have compared the cardiovascular risk between 700 gout patients and 276 controls and reported that risk factors for cardiovascular diseases are disease duration ≥ 2 years, oligo- or poly-arthritis, serum urate level > 0.55 mmol/L at presentation, and joint damage. Philinger et al. [17] have reviewed 1159 gout cases and found that gout patients with cardiovascular diseases are more likely to have obesity, diabetes, osteoarthritis, chronic kidney disease, and prostate disease. Recently published guidelines for gout management also emphasized the importance of regulating traditional risk factors [18]. Therefore, physicians should pay more attention to the control of cardiovascular risk factors in gout patients. 

An interesting result from our study was that hypouricemic agents reduced the risk of CVD. There has been a debate whether urate lowering therapy can reduce the risk of cardiovascular disease in gout patients. A retrospective cohort study from Taiwan has reported that allopurinol has no beneficial effect in preventing cardiovascular outcomes of gout patients [19]. However, that study did not include important risk factors such as smoking, alcohol consumption, BMI, or blood pressure for cardiovascular disease. A meta-analysis of urate lowering therapy on cardiovascular risks has shown that there are no significant differences in cardiovascular events between patients receiving urate lowering therapy and placebo [20]. A recently published meta-analysis has shown opposite results [21]. It included patients with hyperuricemia or gout from seven prospective cohort studies and 17 randomized controlled trials. It argued that cardiovascular events were significantly higher in hyperuricemia patients (relative risk: 1.35, 95% CI: 1.12–1.62). It found that xanthine oxidase inhibitors lowered the risk of cardiovascular events (relative risk: 0.61, 95% CI: 0.44–0.85). Regarding the controversy over the differences in cardiovascular risks of each hypouricemic agents, the CARES study initially stirred up the debate between allopurinol and febuxostat in patients with gout and coexisting cardiovascular disease [22]. The study showed that febuxostat was noninferior to allopurinol in terms of rates of adverse cardiovascular events, all-cause mortality and cardiovascular mortality were higher in patients treated with febuxostat. The following FAST study presented that febuxostat was not associated with an increased risk of cardiovascular outcomes compared with allopurinol [23]. Kang et al. [24] compared cardiovascular risks between allopurinol versus benzbromarone and reported that allopurinol had an increased risk of cardiovascular events and all-cause mortality compared to benzbromarone. Thus, large-scale prospective cohort studies are needed to better address this issue.

The strength of our study was that various covariates were included in the analyses, ranging from lifestyle habits to medications by collecting as much information as we could from the database. However, this study has some limitations. First, the observation period was relatively short. The relatively short follow-up period might have prevented identifying meaningful results. Thus, a sufficient length of observation time is needed in the future to evaluate cardiovascular outcomes. Second, serum uric acid levels were unavailable as an adjustment variable since health insurance claims data were used. Third, we did not compare risk factors of cardiovascular diseases in controls because our aim was to assess a comprehensive list of relevant factors in gout patients. Lastly, despite that our result indicate that cerebrovascular events were reduced in gout patients treated with urate lowering therapy, the specifics (i.e., agent, dose, and duration) were not included in the analysis. 

## 5. Conclusions

Out study showed that the risks of IHD, AMI, and CVD were significantly increased in Korean patients with gout. Old age, current smoking, frequent alcohol intake, high LDL, and diabetes increased the risk of cardiovascular outcomes. However, hypouricemic agents decreased the risk for cerebrovascular diseases. Our data corroborate that it is crucial to identify and manage traditional cardiovascular risk factors besides lowering urate levels in patients with gout.

## Figures and Tables

**Table 1 jcm-11-02124-t001:** Baseline characteristics and outcome events of gout and control patients.

Variable	Gout Patients (*n* = 3306)	Control Patients (*n* = 3306)	*p*-Value
Age (years), mean ± SD	51.64 ± 12.53	51.36 ± 12.17	0.352
Male, *n* (%)	2825 (85.45)	2806 (84.88)	0.511
BMI (kg/m^2^), mean ± SD	25.10 ± 3.21	23.60 ± 2.30	**<0.001 ****
Lifestyle habits			
Smoking, *n* (%)			
Non-smoker	1394 (42.29)	1271 (38.56)	
Past smoker	818 (24.82)	676 (20.51)	**<0.001 ****
Current smoker	1084 (32.89)	1349 (40.93)	
Alcohol intake, *n* (%)			
0–2 days/week	2539 (76.80)	2585 (78.19)	
3–5 days/week	618 (18.69)	554 (16.76)	0.085
≥6 days/week	149 (4.51)	167 (5.05)	
Exercise, *n* (%)			
0–2 days/week	2216 (67.71)	2193 (66.86)	
3–5 days/week	980 (29.94)	1011 (30.82)	0.740
≥6 days/week	77 (2.35)	76 (2.32)	
Laboratory findings			
Fasting glucose (mg/dL), mean ± SD	100.70 ± 23.73	99.39 ± 25.55	**0.027 ***
Total cholesterol (mg/dL), mean ± SD	200.60 ± 40.07	197.80 ± 36.63	**0.003 ****
Triglyceride (mg/dL), mean ± SD	185.20 ± 154.60	146.00 ± 115.60	**<0.001 ****
HDL (mg/dL), mean ± SD	52.46 ± 30.57	54.90 ± 30.64	**0.001 ****
LDL (mg/dL), mean ± SD	117.60 ± 72.12	117.60 ± 57.71	0.999
Creatinine (mg/dL), mean ± SD	1.24 ± 1.42	1.12 ± 1.08	**<0.001 ****
GFR (mL/min), mean ± SD	86.80 ± 35.25	85.38 ± 34.89	0.198
Comorbidities, *n* (%)			
Hypertension	1310 (39.62)	364 (11.01)	**<0.001 ****
Diabetes	659 (19.93)	191 (5.78)	**<0.001 ****
Hyperlipidemia	1328 (40.17)	288 (8.71)	**<0.001 ****
Chronic kidney disease	97 (2.93)	8 (0.24)	**<0.001 ****
Medication, *n* (%)			
Diuretics	434 (13.13)	76 (2.30)	**<0.001 ****
Beta-blocker	339 (10.25)	49 (1.48)	**<0.001 ****
Calcium channel blocker	843 (25.50)	231 (6.99)	**<0.001 ****
ACEI	101 (3.06)	12 (0.36)	**<0.001 ****
ARB	696 (21.05)	180 (5.44)	**<0.001 ****
NSAID	3152 (95.34)	1810 (54.75)	**<0.001 ****
COX-2 inhibitor	86 (2.60)	11 (0.33)	**<0.001 ****
Aspirin	474 (14.34)	93 (2.81)	**<0.001 ****
Steroid	1972 (59.65)	722 (21.84)	**<0.001 ****
Statin	594 (17.97)	125 (3.78)	**<0.001 ****
Colchicine	1555 (47.04)	0 (0)	**<0.001 ****
Hypouricemic agent	1901 (57.50)	1 (0.03)	**<0.001 ****
Outcome events, *n* (%)			
Ischemic heart disease	428 (12.95)	175 (5.29)	**<0.001 ****
Acute myocardial infarction	56 (1.69)	13 (0.39)	**<0.001 ****
Congestive heart failure	123 (3.72)	53 (1.60)	**<0.001 ****
Cerebrovascular disease	137 (4.14)	92 (2.78)	**0.003 ****
Cerebral infarction	119 (3.60)	78 (2.36)	**0.003 ****
Cerebral hemorrhage	18 (0.54)	14 (0.42)	0.502
Transient ischemic attack	72 (2.18)	26 (0.79)	**<0.001 ****
All-cause mortality	80 (2.42)	80 (2.42)	1.000
Cardiovascular mortality	14 (0.42)	12 (0.36)	0.155

BMI, body mass index; HDL, high density lipoprotein; LDL, low density lipoprotein; GFR, glomerular filtration rate; ACEI, angiotensin converting enzyme inhibitor; ARB, angiotensin receptor blocker; NSAID, non-steroidal anti-inflammatory agent; COX, cyclooxygenase; Bold font indicates statistical significance; * *p* < 0.05; ** *p* < 0.01.

**Table 2 jcm-11-02124-t002:** Multivariate Cox proportional HRs for cardiovascular outcomes in gout patients.

Outcome	HR	95% CI	*p*-Value
Ischemic heart disease	1.860	1.446–2.392	**<0.001 ****
Acute myocardial infarction	3.246	1.460–7.217	**0.004 ****
Congestive heart failure	1.399	0.853–2.295	0.183
Cerebrovascular disease	1.552	1.177–2.046	**0.002 ****
Cerebral infarction	1.061	0.687–1.637	0.790
Cerebral hemorrhage	0.779	0.283–2.145	0.629
Transient ischemic attack	1.233	0.651–2.336	0.521
All-cause mortality	1.199	0.737–1.952	0.465
Cardiovascular mortality	1.377	0.452–4.188	0.573

HR, hazard ratio; CI, confidence interval; Bold font indicates statistical significance; ** *p* < 0.01.

**Table 3 jcm-11-02124-t003:** Multivariate Cox proportional HRs for IHD, AMI, and CVD in gout patients.

Variable	IHD		AMI		CVD	
	HR (95% CI)	*p*-Value	HR (95% CI)	*p*-Value	HR (95% CI)	*p*-Value
Age						
20–39			Reference			
40–59	2.142 (1.393–3.294)	**<0.001 ****	3.773 (0.855–16.655)	0.080	3.660 (1.821–7.357)	**<0.001 ****
≥60	3.544 (2.223–5.649)	**<0.001 ****	6.737 (1.405–32.315)	**0.017 ***	8.985 (4.364–18.500)	**<0.001 ****
Sex						
Female			Reference			
Male	0.975 (0.701–1.357)	0.883	0.765 (0.302–1.937)	0.572	0.953 (0.657–1.383)	0.800
BMI (kg/m^2^)						
<18.5	1.418 (0.693–2.900)	0.339	0.000	0.000	0.783 (0.287–2.135)	0.633
18.5–24.9			Reference			
25–29.9	1.097 (0.888–1.354)	0.392	0.950 (0.535–1.686)	0.860	1.119 (0.871–1.439)	0.379
≥30	1.074 (0.713–1.627)	0.733	0.236 (0.031–1.784)	0.162	1.203 (0.721–2.006)	0.480
Lifestyle habits						
Smoking						
Non-smoker			Reference			
Past smoker	1.020 (0.713–1.617)	0.882	0.916 (0.421–1.992)	0.824	0.772 (0.561–1.062)	0.111
Current smoker	1.315 (1.021–1.695)	**0.034 ***	1.581 (0.787–3.174)	0.198	1.054 (0.777–1.429)	0.734
Alcohol intake						
0–2 days/week			Reference			
3–5 days/week	1.144 (0.891–1.467)	0.292	1.174 (0.580–2.376)	0.657	1.343 (0.991–1.819)	0.057
≥6 days/week	0.944 (0.609–1.461)	0.795	1.841 (0.696–4.865)	0.219	1.636 (1.069–2.502)	**0.023 ***
Exercise						
0–2 days/week			Reference			
3–5 days/week	0.862 (0.694–1.072)	0.182	0.750 (0.399–1.411)	0.373	1.099 (0.854–1.414)	0.463
≥6 days/week	1.023 (0.557–1.877)	0.942	0.748 (0.101–5.547)	0.777	1.147 (0.584–2.252)	0.690
Laboratory findings						
Fasting glucose						
<126 mg/dL			Reference			
≥126 mg/dL	0.904 (0.641–1.274)	0.564	0.526 (0.179–1.542)	0.242	0.909 (0.608–1.359)	0.641
Triglyceride						
<150 mg/dL			Reference			
≥150 mg/dL	1.152 (0.935–1.420)	0.184	1.461 (0.810–2.636)	0.208	0.992 (0.771–1.276)	0.948
HDL						
≥40 mg/dL			Reference			
<40 mg/dL	0.882 (0.694–1.120)	0.303	0.876 (0.456–1.685)	0.692	0.788 (0.592–1.051)	0.105
LDL						
<100 mg/dL			Reference			
100–129 mg/dL	1.158 (0.915–1.467)	0.223	0.735 (0.333–1.620)	0.445	1.040 (0.786–1.376)	0.783
130–159 mg/dL	1.146 (0.880–1.493)	0.312	1.763 (0.866–3.591)	0.118	0.960 (0.696–1.324)	0.802
≥160 mg/dL	0.918 (0.632–1.332)	0.652	3.008 (1.368–6.618)	**0.006 ****	0.870 (0.554–1.367)	0.547
GFR						
≥90 mL/min			Reference			
60–89 mL/min	0.933 (0.681–1.279)	0.668	0.443 (0.173–1.134)	0.090	0.956 (0.636–1.435)	0.827
30–59 mL/min	0.766 (0.501–1.171)	0.218	0.880 (0.302–2.563)	0.815	1.185 (0.734–1.914)	0.487
<30 mL/min	0.998 (0.743–1.341)	0.992	0.911 (0.416–1.995)	0.816	1.023 (0.695–1.506)	0.907
Comorbidities						
Hypertension	1.112 (0.811–1.525)	0.510	1.143 (0.472–2.765)	0.767	1.154 (0.804–1.657)	0.437
Diabetes	1.158 (0.912–1.470)	0.228	2.070 (1.118–3.833)	**0.021 ***	1.384 (1.056–1.815)	**0.019 ***
Hyperlipidemia	1.060 (0.825–1.362)	0.648	0.769 (0.389–1.524)	0.452	1.089 (0.818–1.450)	0.558
CKD	1.460 (0.945–2.255)	0.088	1.440 (0.424–4.885)	0.559	1.045 (0.575–1.901)	0.885
Medication						
Diuretics	1.035 (0.795–1.348)	0.798	0.991 (0.480–2.046)	0.981	1.134 (0.839–1.533)	0.413
Beta-blocker	1.301 (0.987–1.715)	0.062	1.132 (0.514–2.495)	0.758	0.977 (0.692–1.378)	0.893
CCB	0.955 (0.728–1.252)	0.738	0.978 (0.467–2.049)	0.954	0.792 (0.581–1.079)	0.140
ACEI	1.132 (0.729–1.756)	0.581	1.023 (0.300–3.489)	0.971	0.890 (0.517–1.531)	0.673
ARB	1.155 (0.893–1.494)	0.273	1.519 (0.738–3.129)	0.256	1.181 (0.872–1.599)	0.283
NSAID	0.956 (0.617–1.482)	0.841	3.171 (0.432–23.284)	0.257	2.050 (0.960–4.377)	0.064
Cox-2 inhibitor	1.067 (0.658–1.731)	0.793	0.282 (0.038–2.096)	0.216	1.225 (0.749–2.002)	0.419
Aspirin	1.208 (0.939–1.555)	0.142	0.825 (0.395–1.723)	0.608	1.244 (0.928–1.668)	0.144
Steroid	1.015 (0.833–1.237)	0.884	0.770 (0.451–1.314)	0.338	1.079 (0.849–1.370)	0.534
Statin	1.355 (0.933–1.578)	0.284	0.966 (0.433–2.155)	0.933	0.971 (0.698–1.350)	0.971
Colchicine	1.146 (0.938–1.401)	0.183	0.732 (0.415–1.290)	0.281	1.202 (0.947–1.525)	0.130
Hypouricemic agent	0.867 (0.704–1.067)	0.177	0.945 (0.534–1.671)	0.846	0.713 (0.556–0.915)	**0.011 ***

HR, hazard ratio; IHD, ischemic heart disease; AMI, acute myocardial infarct; CVD, cerebrovascular disease; BMI, body mass index; HDL, high density lipoprotein; LDL, low density lipoprotein; GFR, glomerular filtration rate; ACEI, angiotensin converting enzyme inhibitor; ARB, angiotensin receptor blocker; NSAID, non-steroidal anti-inflammatory agent; COX, cyclooxygenase; Bold font indicates statistical significance; * *p* < 0.05; ** *p* < 0.01.

## Data Availability

Data are available from the corresponding author upon reasonable request.

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
