# Peer review of "Cardiovascular Risks in Korean Patients with Gout: Analysis Using a National Health Insurance Service Database"

_jcm, 2022, doi:10.3390/jcm11082124_

Round 1

Reviewer 1 Report

The presented manuscript (10 pages, 3 tables, 21 references) reports the data from a case-control study, not from a cohort trial as suggested in the text.

It shows the expected, already many times published results (i.e. gouty patients have more hypertension, more diabetes, more dyslipidemia, more cardiovascular events, taking more drugs, etc.). These persons are no doubt exposed to elevated risk. The surprising finding that they smoke much less, and consume less ethanol, which deserves some comment.

The Cox analysis showed definitely less cerebrovascular (CVD) events among those taking hypouricemic drugs (HR 0.713) with no difference in IHD or MI, which again deserves better argumentation. Moreover, this correlation does not suggest a cause-and-effect phenomenon and does not imply that intake of any hypouricemic drug should be recommended for the prevention of atherosclerotic complications in gouty subjects. Think about Sir Bradford Hill's criteria published over 50 years ago (Proc Roy Soc Med 1965;58:295-300)! Attenuate therefore the bold, presumptuous title!

Author Response

Reviewer1

Point 1. The presented manuscript (10 pages, 3 tables, 21 references) reports the data from a case-control study, not from a cohort trial as suggested in the text.

Response 1: Thank you for the remarks. We understand your concern thus revised the manuscript. However, the Korean National Insurance Service (NHIS) has officially termed the sample data as NHIS-National Sample Cohort, so we left this as it is. We have edited the manuscript as follows:

Line 16:             This retrospective study used sampled clinical data

Line 50:             The NHIS-NSC is a population-based database

Line 61:             we conducted a retrospective study

Line 144:           We utilized the NHIS-NSC and

Line 174:           from the NHIS-NSC database

Point 2. It shows the expected, already many times published results (i.e. gouty patients have more hypertension, more diabetes, more dyslipidemia, more cardiovascular events, taking more drugs, etc.). These persons are no doubt exposed to elevated risk. The surprising finding that they smoke much less, and consume less ethanol, which deserves some comment.

Response 2: Thank you for sharing your insights. The data shows that the percentage of smoking between the two groups are statistically different, yet not drastically. Nevertheless, the percentage is indeed different. We assume that the cross-sectional data suggest that patients with gout, having higher prevalence of hypertension, diabetes, dyslipidemia, and cardiovascular events, etc., are maintaining a healthier lifestyle or decided to quit smoking. This is shown in higher percentage of past smokers in patients with gout. We added the following in Discussion:

It is noteworthy that baseline characteristics showed that patients with gout smoked less and consumed less alcohol. As they had a higher prevalence of hypertension, diabetes, dyslipidemia, and cardiovascular events, patients may have had chosen to maintain a healthier lifestyle and decided to quit smoking; the percentage of past smokers was higher in patients with gout.

Point 3. The Cox analysis showed definitely less cerebrovascular (CVD) events among those taking hypouricemic drugs (HR 0.713) with no difference in IHD or MI, which again deserves better argumentation. Moreover, this correlation does not suggest a cause-and-effect phenomenon and does not imply that intake of any hypouricemic drug should be recommended for the prevention of atherosclerotic complications in gouty subjects. Think about Sir Bradford Hill's criteria published over 50 years ago (Proc Roy Soc Med 1965;58:295-300)! Attenuate therefore the bold, presumptuous title!

Response 3: We think that these points are valid and important. We were alarmed on identifying the lower risk or cerebrovascular disease in users with urate lowering therapy in our population, yet again agree that this would need to be interpreted with caution. In consideration to the comments from Reviewer1 (and Reviewer2), we revised the title of the manuscript to ‘Cardiovascular risks in Korean patients with gout: analysis using a National Health Insurance Service database’.

Reviewer 2 Report

This is a very interesting study about the assessment of cardiovascular disease and mortality associated in a population of Korean gout patients. This is a retrospective study, with a follow-up period of 5 years.

The study population is composed of patients with early onset gout with a duration < 5 years (if I am wrong, please specify the gout duration).

The authors found an increased risk of ischemic heart disease, acute myocardial infarction and cerebrovascular diseases in gout patient. They have identified associated risk factors such as age, smoking status, diabetes and alcohol. They found a reduction of cerebrovascular events in patient using ULT.

Major comments:

  • Why the authors did not follow the patients over a longer period (no data available after 2015 ?)
  • The lack of serum urate level is a major limitation. Searching for an association between serum urate level and cardiovascular events would have been particularly interesting.
  • How do the authors explain the absence of cardiovascular events associated with NSAID and corticosteroid use? Are there any data about cumulated doses and duration of use?

  • About table 1: how do the authors explain the difference in cardiovascular events between gout patients and control patients? Indeed, gout patients have more comorbidities, hyperlipidemia, etc…

About Urate lowering therapies:

  • It would have been interesting to have the proportion of patient using allopurinol, febuxostat and benzbromarone & analysis about cardiovascular event according to ULT (allopurinol vs febuxostat vs benzbromarone).
  • It will provide more information about the reduced risk of cerebrovascular events in this population. Indeed, there are some contradictory results about cardiovascular risk in patient using ULT (for example, Kang et al., from a Korean gout population, found an increase in CV events in patients treated with allopurinol compared to benzbromarone).
  • Furthermore, there is still some doubt about the increased cardiovascular risk in patient treated with febuxostat.
  • Similarly, were the authors able to access the ULT duration? If they do, please provide more details.
  •  
  • About table 2: Multivariate analysis compare gout patients versus controls? If so, I would add in the results section "in comparison to control patient after adjusting..." L117
  •  
  • About the title: in view of the limitations related to the ULT analysis, the lack of details about ULT, the title of the study seems not appropriate. Please change for a more general title about the comorbidities associated with cardiovascular events in Korean patients with gout disease.

Discussion:

  • Please provide more information about contradictory results regarding the cardiovascular risk in patient with ULT
  • kang et DOI: 10.1093/eurheartj/ehab619, FAST study, DOI:10.1016/S0140-6736(20)32234-0)
  • Please discuss about the limitations of the results concerning the reduction of the cerebrovascular events and in particular the lack of details about ULT ((unless the authors can perform new analysis with ULT details)

Minor comments :

  • Table 2 is difficult to read, please correct,
  • Maybe embold significant results in table 3.

Author Response

Reviewer2

Point 1. This is a very interesting study about the assessment of cardiovascular disease and mortality associated in a population of Korean gout patients. This is a retrospective study, with a follow-up period of 5 years. The study population is composed of patients with early onset gout with a duration < 5 years (if I am wrong, please specify the gout duration).

The authors found an increased risk of ischemic heart disease, acute myocardial infarction and cerebrovascular diseases in gout patient. They have identified associated risk factors such as age, smoking status, diabetes and alcohol. They found a reduction of cerebrovascular events in patient using ULT.

Response 1: Thank you for your remarks. We applied an operational definition aiming to enroll ‘new’ cases of gout. Patients with gout were to have claim data with an ICD-10 code M10 between January 1, 2009, and December 31, 2010: patients diagnosed with gout in 2008 were excluded. As such, this was a retrospective study using a claim database, so we could not state the exact disease duration of the study subjects.

Major comments:

Point 2. Why the authors did not follow the patients over a longer period (no data available after 2015?)

Response 2: The NHIS-NSC database encloses data between 2002 and 2015. We wanted to include laboratory findings as covariates, and some important results such as creatinine and lipid profiles were not available until 2009. As lipid profiles are crucial risk factors for cardiovascular diseases, we determined to enroll patients from 2009. These are described in the study population section of Methods.

Point 3. The lack of serum urate level is a major limitation. Searching for an association between serum urate level and cardiovascular events would have been particularly interesting.

Response 3: Thank you for your insightful comment. We completely agree with this, however, serum urate levels unfortunately were not provided in the NHIS-NSC database. We plan to study this aspect by using data from a gout registry that our group has established.

Point 4. How do the authors explain the absence of cardiovascular events associated with NSAID and corticosteroid use? Are there any data about cumulated doses and duration of use?

Response 4: Thank you for this interesting question. Unlike the previous studies studying patients on daily NSAIDs or corticosteroids, most patients with gout use these sporadically or intermittently, thus medications may have had less effect on cardiovascular outcomes. In light of your second question, we did not initially plan to obtain the cumulated dose and duration of use of these medications for the aforementioned reason.

Point 5. About table 1: how do the authors explain the difference in cardiovascular events between gout patients and control patients? Indeed, gout patients have more comorbidities, hyperlipidemia, etc…

Response 5: First, we sincerely apologize that we made a mistake in applying the results of ‘outcome events’ in Table 1; the results of gout patients and control patients were misplaced. This was corrected as we revised our paper and double checked that there is no other error. We the authors are again truly sorry for this. The debate has been ongoing whether gout itself is an independent risk factor for cardiovascular (CV) events or comorbidities increase the CV risk. As we mentioned in Discussion, Huang et al. reported cumulative incidence of acute myocardial infarction was significantly higher in patients with gout even after propensity score matching of comorbidities (J Investig Med 2021, 69, 1161-7). In addition, hyperuricemia itself ensues oxidative stress leading to a systemic inflammatory response and atherosclerosis. We understand certain comorbidities can contribute to developing CV disease. Nevertheless, we consider gout itself to be an important risk factor for CV diseases.

Point 6. About Urate lowering therapies: It would have been interesting to have the proportion of patient using allopurinol, febuxostat and benzbromarone & analysis about cardiovascular event according to ULT (allopurinol vs febuxostat vs benzbromarone). It will provide more information about the reduced risk of cerebrovascular events in this population. Indeed, there are some contradictory results about cardiovascular risk in patient using ULT (for example, Kang et al., from a Korean gout population, found an increase in CV events in patients treated with allopurinol compared to benzbromarone). Furthermore, there is still some doubt about the increased cardiovascular risk in patient treated with febuxostat. Similarly, were the authors able to access the ULT duration? If they do, please provide more details.

Response 6: Thank you for your insightful remarks. First, we were unable to analyze the duration of urate lowering therapy. We are aware of the publications of Kang et al., regarding not only allopurinol versus benzbromarone (Eur Heart J 2021, 42:4578-88), but also allopurinol versus febuxostat (Rheumatology 2019, 58:2122-29) looking into cardiovascular (CV) events in Koreans, which both used the National Health Insurance Service database. Along with the CARES and FAST trial, the CV risk in users of the three agents are intertwined depending on the study population, specific outcomes, and comparators. Our group is aiming to study the CV risks in users of respective agents in a gout registry that is underway. Overall, our study was unable to dissect the risks in respective urate lowering agents yet presented that the risk of cerebrovascular disease was reduced in patients treated with urate lowering therapy.

Point 7. About table 2: Multivariate analysis compare gout patients versus controls? If so, I would add in the results section "in comparison to control patient after adjusting..." L117

Response 7: Thank you for this comment. We added the sentence as you recommended in Results.

Point 8. About the title: in view of the limitations related to the ULT analysis, the lack of details about ULT, the title of the study seems not appropriate. Please change for a more general title about the comorbidities associated with cardiovascular events in Korean patients with gout disease.

Response 8: Thank you for your remarks. In consideration to the comments from Reviewer2 (and Reviewer1) we revised the title of the manuscript to ‘Cardiovascular risks in Korean patients with gout: analysis using a National Health Insurance Service database’.

Discussion:

Point 9. Please provide more information about contradictory results regarding the cardiovascular risk in patient with ULT

kang et DOI: 10.1093/eurheartj/ehab619, FAST study, DOI:10.1016/S0140-6736(20)32234-0)

Response 10: First, we would like to state that our aim was to compare the outcomes in patients that were on urate lowering therapy versus those who were not. As for the mentioned papers, they compared the outcomes in users of different urate lowering agents, yet lacked a control group (i.e, not using urate lowering therapy), which was an important limitation. Therefore, we do not regard our results to be contradictory. On the other hand, we also understood that a description of these studies are their implications should be included in the manuscript. Therefore, we added the following in Discussion:

Regarding the controversy over the differences in cardiovascular risks of each hypouricemic agents, the CARES study initially stirred up the debate between allopurinol and febuxostat in patients with gout and coexisting cardiovascular disease (N Engl J Med 2018, 378, 1200-1210). The study showed that febuxostat was noninferior to allopurinol in terms of rates of adverse cardiovascular events, all-cause mortality and cardiovascular mortality were higher in patients treated with febuxostat. The following FAST study presented that febuxostat was not associated with an increased risk of cardiovascular outcomes compared with allopurinol ( The Lancet 2020, 396, 1745-1757). Kang et al (Eur Heart J 2021, 42, 4578-4588) compared cardiovascular risks between allopurinol versus benzbromarone and reported that allopurinol had an increased risk of cardiovascular events and all-cause mortality compared to benzbromarone.

Point 11. Please discuss about the limitations of the results concerning the reduction of the cerebrovascular events and in particular the lack of details about ULT ((unless the authors can perform new analysis with ULT details)

Response 11: Thank you for your suggestion. We added the following to Discussion:

Lastly, despite that our result indicate that cerebrovascular events were reduced in gout patients treated with urate lowering therapy, the specifics (i.e., agent, dose, and duration) were not included in the analysis.

Minor comments :

Point 12. Table 2 is difficult to read, please correct,

Response 12: We edited table 2 as recommended.

Point 13. Maybe embold significant results in table 3.

Response 13: We edited table 3 as recommended. Significant values in other tables are also indicated in bold fonts for the sake of unity.

Round 2

Reviewer 2 Report

No further comments.

The authors have responded to all comments. The corrections of errors in table 1make the results more understandable.

Author Response

Point 1. No further comments.

The authors have responded to all comments. The corrections of errors in table 1make the results more understandable.

Response 1: We appreciate your comment. Thanks to your remarks, we believe the quality of this paper has improved. We once again apologize for the error made in Table 1.